# Transcutaneous spinal direct current stimulation (tsDCS) does not affect postural sway of young and healthy subjects during quiet upright standing

**Felipe Fava de Lima** *, **Cristiano Rocha Silva, Andre Fabio Kohn**

Biomedical Engineering Laboratory, Escola Politécnica, University of São Paulo, São Paulo, Brazil

* felipefavadelima@usp.br

**Data Availability Statement:** All data files are available from the OSF database (osf.io/c24ta) DOI:10.17605/OSF.IO/C24TA.

## Abstract

Transcutaneous spinal direct current stimulation (tsDCS) is an effective non-invasive spinal cord electrical stimulation technique to induce neuromodulation of local and distal neural circuits of the central nervous system (CNS). Applied to the spinal cord lumbosacral region, tsDCS changes electrophysiological responses of the motor, proprioceptive and nociceptive pathways, alters the performance of some lower limb motor tasks and can even modulate the behavior of supramedullary neuronal networks. In this study an experimental protocol was conducted to verify if tsDCS (5 mA, 20 minutes) of two different polarizations, applied over the lumbosacral region (tenth thoracic vertebrae (T10)), can induce changes in postural sway oscillations of young healthy individuals during quiet standing. A novel initialization of the electrical stimulation was developed to improve subject blinding to the different stimulus conditions including the sham trials. Measures of postural sway, both global and structural, were computed before, during and following the DC stimulation period. The results indicated that, for the adopted conditions, tsDCS did not induce statistically significant changes in postural sway of young healthy individuals during quiet standing.

## Introduction

Transcutaneous spinal direct current stimulation (tsDCS) is a relatively new non-invasive technique able to induce long-lasting neuromodulation on the human central nervous system (CNS) [1]. It is performed through a transcutaneous electrode placed over the human torso at the back of the spine and a second electrode placed elsewhere on the torso so that a low intensity direct electric current (DC) flows through the spinal cord during a predefined time interval (usually more than 15 minutes) [2].

Depending on the polarity and electrode placement, direct current (DC) has been shown to induce changes on lower limb somatosensory evoked potentials [1], alter the synaptic efficacy of the lumbar spinal monosynaptic reflex circuit formed by Ia afferents and motoneurons [3–7], modify spinal nociceptive circuit gains [8–11], improve motor unit recruitment [12],

**Funding:** São Paulo Research Foundation (FAPESP grant #2016/10614-4 from CRS) (https://fapesp.br/) Brazilian Council of Science and Technology (CNPq grant # 311223/2021-4 from AFK) (https://www.gov.br/cnpq/pt-br) The funders had no role in study design, data collection and analysis, decision to publish, or preparation of the manuscript.

**Competing interests:** The authors have declared that no competing interests exist.

reduce presynaptic D1 inhibition possibly by neuromodulation of spinal interneurons [13], change corticospinal transmission/excitability [14, 15] and facilitate TA muscle proprioceptive transcortical reflexes [16]. Interestingly, in addition to the segmental effects generated at the stimulated spinal cord region, tsDCS may induce neuromodulation of supramedullary and cortical neural circuits. Some studies have shown changes of intracortical inhibition/facilitation [17, 18], effects on cerebellar-cortical neuronal networks [19] and modulation of the interhemispheric processing delay [20] due to tsDCS applied over lower thoracic regions. The DC stimulation can also induce alterations in central fatigue mechanisms [21] and improve locomotor learning tasks [22].

Similarly to transcranial direct current stimulation (tDCS), the physiological mechanisms underlying the effects of tsDCS are not yet completely understood [23]. The electric field produced by the electric current flow through the spinal cord possibly induces a slight polarization shift of some spinal cord neuronal compartments, mainly axonal terminals [24], changing the neuronal excitability and thus modifying the firing behavior of the stimulated cells. Over time these changes can produce persistent modifications of L-VGCC (L-Type Voltage Gated Calcium Channel) ionic channels or/and NMDA (N-methyl-D-aspartate) receptors [25, 26] (a review on tDCS is provided by [23]). Other mechanisms can also contribute to the observed effects, as for example, ionic channel migration, electrical stimulation of non-neural cells, changes in neurotransmitter concentrations, and/or nano-galvanotropism [23].

The electric field profile generated in the spinal cord by tsDCS depends not only on the position of the electrode located over the spinal cord region, but also on where the other electrodes are attached. Previous computational studies using realistic human torso models have shown that the field generated in the region of the lumbosacral enlargement can be maximized by placing one electrode over the tenth thoracic vertebra (T10) and electrodes over the iliac crests [24, 27].

Postural control requires the CNS to integrate sensory information from the visual, auditory, vestibular and somatosensory systems for the proper activation of the skeletal muscles that are involved in balance control [28, 29], assigning different weights to the information from each system in different scenarios [28, 30]. The quiet standing posture is commonly used to study aspects of the behavior of the postural control system [31–35] by analysing the resultant postural oscillations [36]. Commonly, a force platform is used to acquire the position of the center of pressure (COP) over time, which corresponds to the location of the resulting ground force reaction generated by the feet over time. Quantification of postural oscillations during quiet standing can be based on parameters obtained in the time domain, either by simple measures such as the standard deviation [37, 38] or by more complex ones such as entropy [39, 40] and fractal structure [41], or in the frequency domain [37]. Different parameters can capture different aspects of the underlying neural control during quiet standing [42] and there is no consensus on which parameters should be used to identify changes due to different experimental conditions or due to disease [36, 43].

In healthy subjects, during quiet standing, the CNS mainly uses the ankle strategy to maintain balance [44], activating leg muscles in response to sensory input from lower limb muscle spindles and Golgi tendon organs (GTO) and from cutaneous mechanoreceptors of the soles of the feet [45–49]. The neural circuitry of the spinal cord certainly has a key role in the control of many complex motor tasks [50]. A multiscale mathematical model study showed that a pattern of activation of the leg muscles similar to that obtained in experimental studies during standing posture, could be obtained just with local spinal cord circuitry without longer feedback from supraspinal circuits [51]. Another link between postural oscillations and spinal neural circuits is suggested by the modulation of the triceps surae Hoffman reflex (H-reflex) according to the postural sway phase and direction during quiet standing [52–54], possibly by

a modulation of presynaptic inhibition [55]. Another study revealed that sublimiar electrical noise stimulation was sufficient to alter characteristics of the postural sway during quiet standing posture [56], probably by a stochastic resonance mechanism acting on muscle splindles.

Through the extrapolation of data obtained from cats, the activity of each human lumbar motoneuron is probably influenced by tens of thousands of axonal terminations [57]. These come from a large number of axons originating from cortical and subcortical supramedullary nuclei centers as well as from spinal cord interneurons responsible for the integration of information from other motorneurons and from proprioceptive inputs carried by type Ia, Ib and II afferent axons [58–60]. Each motoneuron also receives direct excitatory connections from type Ia fibers originating from muscle spindles. Most inputs to a given motoneuron can act both on ionotropic and metabotropic receptors. The latter are responsible for persistent inward currents (PICs) that produce longer lasting changes in motoneuron excitability [61]. All the axonal terminals acting on spinal cord motoneurons and interneurons are susceptible to undergo alterations induced by an electrical field generated by tsDCS [24]. Therefore, if the DC current is targeted on the lumbar region of the spinal cord, axonal endings involved in motor control may undergo changes that influence the control of upright standing.

Summarizing, the general hypothesis of the present study was that tsDCS applied over the lumbosacral enlargement can modify the behavior of postural sway of young and healthy individuals during quiet standing. This hypothesis is based on several published data that suggest that both segmental and suprasegmental regions of the central nervous system contribute to motor control during upright posture [47, 48, 62–65]. These regions could be affected both by segmental effects of tsDCS, such as changes on spinal reflexes, and/or by supramedullary effects, as for example by the neuromodulation of axonal endings of first-order neurons of spinocerebellar pathwayss related to postural control [66, 67]. In order to test this hypothesis, we conducted an experimental protocol applying lumbar-level tsDCS with two different polarizations and analyzed the postural sway by posturography before, during and after of the electrical stimulation. An experimental blinding assessment methodology and a novel initialization of the electrical stimulation was also developed. To our knowledge this is the first study that attempts to verify the effect of tsDCS on quiet standing.

## Materials and methods

### Subjects

Seventeen young healthy volunteers were enrolled in the study. All participants reported having no previous history of orthopedic injuries, diseases of the nervous system, chronic pain, labyrinthitis or diabetes and had no knowledge of effects of electrical stimulation in humans. On experiment days the participants did not take stimulant substances, such as coffee or energy drinks, and did not take any medications other than those they were already used to. The study was approved by the Research Ethics Committee of the Physical Education School of the University of São Paulo (CAAE 09592919.0.0000.5391) and all participants read and signed the approved free and informed consent term (TCLE).

As no previous studies of similar scope were found, the sample space size was defined based on values adopted in posturography studies in the literature during quiet standing as well as in studies on tsDCS. Typical sample size in these fields have been between 10 and 20 participants.

### tsDCS protocol

The DC stimulation was generated by a commercial electrical stimulator (Stmisol-1, Biopac System, Inc., EUA) connected to three rectangular self-adhesive disposable electrodes (10x5 cm$^2$, ValuTrode VL4595, Axelgaard Manufacturing CO. Ltd., EUA). To maximize the electric

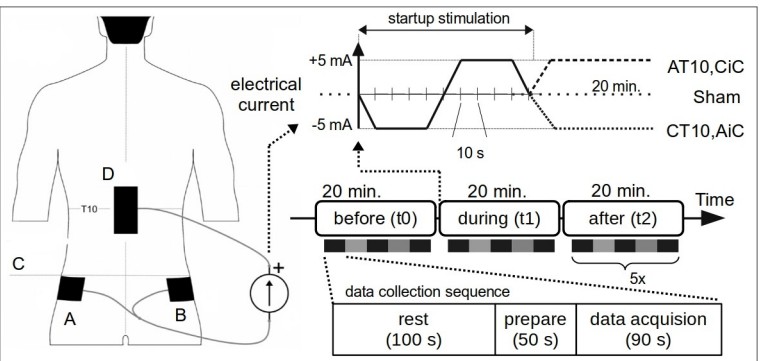

**Fig 1. Electrode placement and experimental protocol.** Back torso illustration on the left: One electrode (10x5 cm$^2$) was placed over T10 (D) and two electrodes (10x5 cm$^2$) on the upper edge (C) of the iliac crests (A and B). The three protocols were composed of a startup stimulation (upper right plot) consisting of a 5 mA 50-seconds stimulation with the electrode over the spinal cord configured as cathode followed by a 5 mA 50-seconds stimulation of opposite polarity. On each experimental day, 5 repetitions of a data collection sequence (resting in a chair for 100 seconds, 50 seconds to prepare and start the quiet standing task and 90 seconds of data acquisition, as shown by the bottom right scheme) were performed both before (t0), during (t1) and after (t2) the electrical stimulation protocol (center right scheme). During the electrical stimulation protocol (t1), after the startup stimulation, the stimulator was held either at +5 mA (AT10,Cic), -5mA (CT10,Cic) or turned off (Sham) for 20 minutes. The electrical slew rate was 0.5 mA/s. The electrical stimulator was turned off before (t0) and after (t2) the electrical stimulation protocol.

field intensity generated by tsDCS in the region of lumbosacral enlargement [24, 27], one electrode was placed centralized over T10, with the largest dimension oriented in the rostrocaudal direction, and two electrodes were positioned over the upper edge of the iliac crests on each side of the body, with the largest dimension positioned in the anteroposterior direction (Fig 1, left illustration). The electrode placed over T10 was connected to one terminal of the electrical stimulator, and the two electrodes positioned over the iliac crests were connected together to the other terminal. The skin below the contact region was previously gently cleaned with cotton moistened with alcohol 70%. T10 was identified by palpation with the supervision of a physical therapist (the second author).

The electrical stimulation was performed with an intensity of 5 mA during 20 minutes, resulting in a maximum electrical current density at the skin-electrode interface of 100 uA/cm$^2$ and a delivered electrical charge of 120 mC/cm$^2$. These values are well below tissue damage [68–70]. The adopted current density is slightly higher than the adopted by some authors (71.5 uA/cm$^2$) [1, 4, 8, 9, 12, 17, 71, 72], it is the same as that used in one study [73] and it is four times lower than that employed in another one (400 uA/cm$^2$) [18]. During the entire protocol, the electrical current applied to the participant was monitored by a calibrated battery-powered ammeter (MD5880, Icel, Brazil).

Three electric stimulation protocols were investigated: Cathode over T10 and anode over the iliac crests (CT10,AiC); Anode over T10 and cathode over the iliac crests (AT10,CiC); Sham. To improve experimental blinding and to try to diminish any possible effects of the electrical stimulation during the Sham protocol [74], all three stimulation protocols were initialized with an electrical start-up stimulation consisting of a 5 mA 50-seconds stimulation with the electrode over the spinal cord configured as cathode followed by a 5 mA 50-seconds stimulation of opposite polarity (Fig 1, upper plot). In the Sham protocol, after the application of the startup stimulation, the electrical stimulator was turned off. All electric current changes occurred at a maximum rate of 0.5 mA/s, minimizing skin discomfort and muscular activation.

## Posturography by force platform

The resulting reaction forces (Fx, Fy and Fz) and moments (Mx, My and Mz) generated by the feet of the participants on the ground during the quiet standing posture were measured by a calibrated force plate (OR6–7-1000, AMTI Advanced Mechanical Technology, Inc., EUA). Participants were positioned on bipedal standing at the center of the force platform, barefoot, feet positioned at a comfortable distance less than shoulder-width apart. Markings were made on the force platform so that the positioning of the feet could be replicated throughout the experiment. During data acquisition, subjects were instructed to remain as still as possible, with arms relaxed at their sides, with eyes closed and covered by opaque glasses and ears covered by headphones reproducing white noise at a pleasant volume.

## Experimental procedure

The experimental procedure was performed according to the Helsinki declaration [75]. Participants attended the laboratory three times with a minimum interval of three days between experimental sessions. No previous information about the electrical stimulation characteristics was provided to the participants. On each day, the subject performed the quiet standing protocol in one of the three different electrical stimulation protocols: (CT10, AiC), (AT10, CiC) or Sham, chosen at random and prioritizing the appropriate balance of the selected sequence of electrical stimulation protocols between the participants.

On each experimental day, after the participant's preparation, seventeen repetitions of a data collection sequence were performed both before (t0), during (t1) and after (t2) the electrical stimulation protocol (Fig 1, center right) as follows: In order to familiarize the subject with the experimental procedure, the experimental protocol (i.e., postural sway acquisition in quiet stance) was initiated by two dummy repetitions. Then five protocol repetitions before (t0), five during (t1) and five after (t2) the electrical stimulation protocol were performed, lasting 60 minutes in total (20 minutes in each of the three stages with 5 repetitions in each). The data collection sequence was composed of three periods: resting, preparation and data acquisition. During the resting period, participants were asked to sit relaxed in a chair for 100 seconds, minimizing possible fatigue effects. In the preparation phase, within 50 seconds, the subjects got up from the chair, positioned their feet on the force platform according to the markings, covered their eyes with opaque glasses, closed their eyes and remained as still as possible in the upright posture with masking white noise sound applied through headphones. During the data acquisition period, 90 seconds of data were obtained from the force platform while the participant was performing the quiet standing task. After this period, participants were instructed to sit back in the chair and wait for a new data acquisition sequence repetition.

The experimental protocol was fully automated by a software developed in Labview (National Instruments, USA) that managed data acquisition, instructed the participant by recorded voice commands and triggered the electrical stimulation protocol, ensuring precise protocol timing.

At the end of each experimental session participants were asked to fill out an assessment form. The subjects had to rate on a five-level scale the duration of the electrical stimulation (level 0 corresponding to no perceived electrical stimulation, level 1 to a very short felt electrical stimulation duration, level 2 to a short one, level 3 to a long duration and level 4 to a very long duration) and on a four-level scale the perceived levels of itching, pain, burning, heating and tingling from the electrical stimulation (level 0 corresponding to no perceived sensation, level 1 to a soft sensation, level 2 to a moderate one and level 3 to an intense sensation). In this form, minor and major adverse events could also be registered by the researcher conducting

the experiment. This assessment methodology is similar to the one used for brain tDCs to verify the effectiveness of experimental blinding and the occurrence of adverse events [76]. A translated version of the assessment form is available on S1 File.

## Signal acquisition and processing

The analog signals from the force plate were digitized by an analog to digital converter (Power1401, Cambridge Electronic Design Limited, UK), with a sampling frequency of 1kHz. The acquired digital data were processed offline in Matlab (Matlab 2015a, MathWorks, EUA). The COP signals, both in the anterior-posterior (AP) and medial-lateral (ML) directions, were obtained from the resulting forces and moments acquired from the force platform [77, 78]. All signals were filtered by a fourth order low-pass digital Butterworth filter with a cutoff frequency of 10 Hz and then resampled to 100 Hz. The first 160 ms of the filtered signals were discarded to eliminate the filter transient period.

The raw COP data in the anteroposterior direction of all subjects were visually inspected by a trained technician. Data acquisitions with anomalous dynamic periods were manually excluded from the data analysis (Fig 2). These periods corresponded to undesirable small movements of the participant generated by a decrease in attention during task execution, deep breathing or small arm movements. Subjects who had more than three excluded data acquisitions at a same stage of the electrical stimulation protocol were removed from the final results.

A set of most commonly adopted COP parameters, both global and structural, were computed by Matlab routines. The following time domain parameters were obtained: standard deviation ($SD$ parameters), mean velocity ($MVELO$ parameters), ellipse area that encompasses the stabilogram area with 95% confidence ($AREAE$ parameters) and sway-area rate ($AREA_{rate}$) [37]. The following frequency domain quantifiers were also computed: power spectral density (PSD) area between 0.05 Hz to 0.5 Hz ($PSD\_AREA\_LF$ parameters), PSD area between 0.5 Hz to 2 Hz ($PSD\_AREA\_HF$ parameters) [37] and the frequencies encompassing 50% ($f50p\_PSD$ parameters) and 80% ($f50p\_PSD$ parameters) of the PSD [37]. To characterize temporal patterns, stabilogram diffusion analysis (SDA) ($D_S$, $D_L$, $\Delta T_c$ and $\langle \Delta X^2_{COP} \rangle$ parameters) [41], frequency-specific fractal analysis (FsFA) ($\alpha\_S$ and $\alpha\_L$ parameters) [79] and multi-scale entropy analysis (MSE) ($CI$ parameters) [80, 81] were also computed. The $SD$, $MVELO$, $f50p\_PSD$, $f80p\_PSD$ and $CI$ parameters were also calculated for the COP velocity and for the signals obtained by the rambling/trembling decomposition as they cover different aspects of the underlying neural control [36, 81, 82].

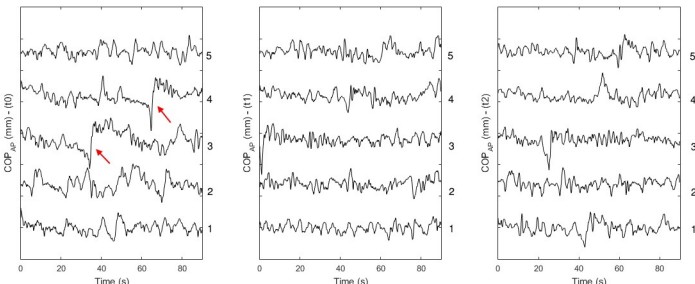

**Fig 2. Data acquisition visual exclusion criteria.** Example of COPs signal in the anteroposterior direction acquired from a subject in one of the electrical stimulation protocols. From left to right: acquisitions obtained before (t0), during (t1) and after (t2) the electrical stimulation protocol. In (t0), acquisitions 3 and 4 presented periods of expressive anomalous dynamics (red arrows), and were excluded from the final analysis.

## Data analyses

Statistically significant differences in the COP parameters were verified by a two-way analysis of variance (ANOVA) using the IBM SPSS Statistics 20 package (IBM, USA) with a significance level of $p \leq 0.05$. The two-way ANOVA main factors were "electrical stimulation protocol" ((CT10, AiC), (AT10, CiC) and Sham) and "electrical stimulation stage" (t0, t1, t2). For those cases where sphericity condition was violated ($p \leq 0.05$), the Greenhouse-Geisser degrees of freedom correction was used for $\epsilon \leq 0.75$ and the Huynh-Feldt correction otherwise [83]. Parameters for which the null hypothesis was rejected, *post hoc* analysis was conducted using the Bonferroni procedure to identify the conditions that showed statistically significant differences [83].

Statistically significant differences in the participants' assessment of the electrical stimulation for different protocols were verified by Friedman tests ($p \leq 0.05$) with independent variable "electrical stimulation protocol" and dependent variables: "duration of electrical stimulation", "itching", "pain", "burning", "heating", and "tingling". In situations where the null hypothesis was rejected, a *post hoc* analysis with the Wilcoxon signed-rank test ($p \leq 0.05$) with Bonferroni correction would be conducted. This analyses was conducted on Python 3.8 with the library SciPy 1.8.0.

## Results

Four subjects were excluded from the final data analysis since they had more than three excluded data acquisitions in a same stage of the electrical stimulation protocol. The results of the COP parameters were obtained from thirteen subjects (5 females; age 23.0 ± 3.5; weight 66.5 ± 10.7 kg; height 1.69 ± 0.08 m [mean ± standard deviation)]). The balance of sequences of the electrical stimulation protocol is available in S2 File. Assessments of the electrical stimulation of twelve participants were used for the evaluation of experimental blinding.

The averages and standard deviations of all COP parameters at different stages (t0, t1 and t2) of the three electrical stimulation protocols, the quartile ranges of the electrical stimulation assessments answers and the results of the statistical analysis, both for the COP parameters and for the results of the assessment of electrical stimulation, are available on the S2 File.

A graphical representation of the obtained results of some parameters, representing a sample of the main COP analyses (time, frequency, SDA, FsFA and MSE), are exhibited in Fig 3 (panels "a" to "h").

Two parameters ($f50p\_PSD_{AP}$ and $\alpha L_{ML}$) out of 34 had statistically significant differences but not associated with the electrical stimulation protocol. Therefore, the hypothesis that tsDCS alters the dynamics of postural control during quiet standing posture on young and healthy subjects has been rejected.

The statistically significant difference observed in $f50p\_PSD_{AP}$ ($p < 0.05$) was in the interaction between stimulation protocol and electrical stimulation stage. *Post-hoc* analysis indicated that a statistically significant difference ($p < 0.05$) was observed before (t0) the electrical stimulation between Sham and (A-T10,CiC) protocols (Fig 3, panel "c").

The $\alpha L_{ML}$ parameter had a statistically significant difference ($p < 0.01$) in the main factor "electrical stimulation stage". *Post-hoc* analysis for this parameter indicated that the null hypothesis was rejected ($p < 0.001$) between during (t1) and after (t2) the electrical stimulation stage (Fig 3, panel "b").

A graphical representation of the participants' assessments of the electrical stimulation protocol is shown in Fig 4. No statistically significant differences were obtained between different electrical stimulation protocols indicating satisfactory experimental blinding. No major adverse effects were reported. All participants presented temporary mild erythema under the

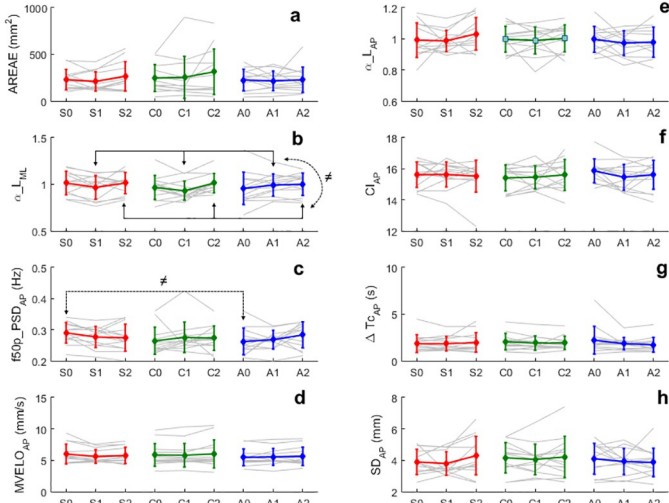

**Fig 3. Graphical representation of some of the results obtained from COP parameters.** Points in red correspond to the average of the parameter, from all participants, before (S0), during (S1) and after (S2) the electrical stimulation protocol in the Sham protocol. Points in green correspond to the average of the parameter, of all participants, before (C0), during (C1) and after (C2) the electrical stimulation protocol in the (CT10, AiC) protocol. Points in blue indicate the parameter average, of all participants, before (A0), during (A1) and after (A2) the electrical stimulation protocol in the (AT10, CiC) protocol. Vertical lines represent the standard deviation of the sample space. Light gray lines represent the parameter values calculated for each subject. $f50p\_PSD_{AP}$ (panel "c") has statistically significant difference before (t0) the electrical stimulation between the Sham and (A-T10,CiC) protocols. $\alpha L_{ML}$ (panel "b") has statistically significant difference between during (t1) and after (t2) the electrical stimulation stage. The "$\neq$" symbol indicates the situations where statistically significant differences were observed.

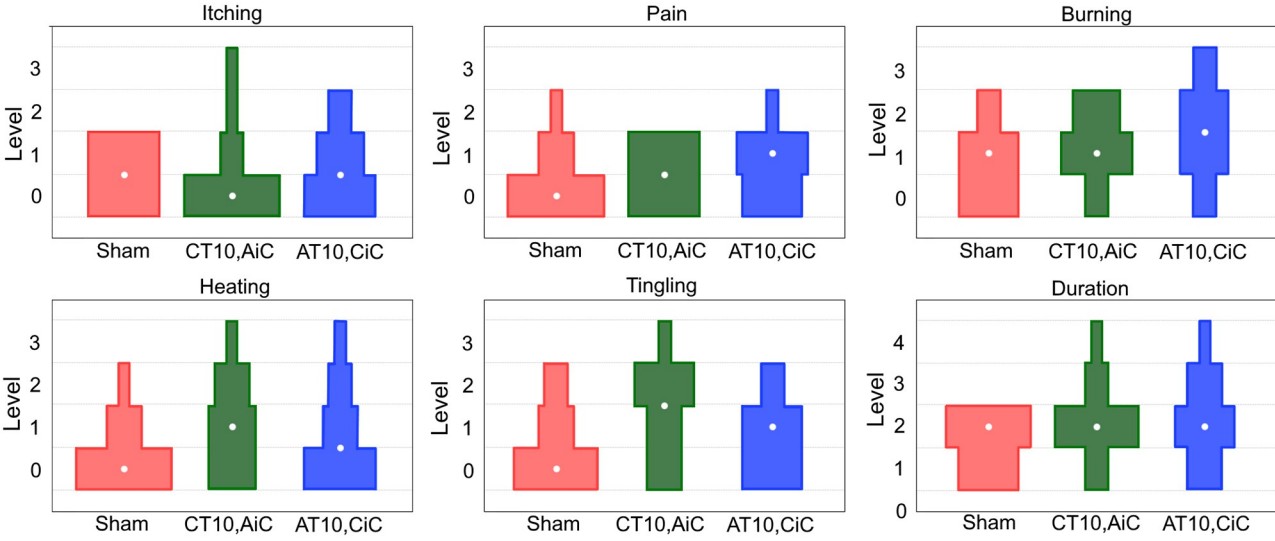

**Fig 4. Graphical representation of the answers from the participants' assessments.** Clockwise, starting from the graph on the left side of the upper corner, are displayed the frequencies of the answers obtained from the electrical stimulation assessments regarding the level of itching, pain, burning, heating, tingling, and duration of the electrical stimulation. The area of the geometric shapes are proportional to the frequency of answers obtained in each condition. For example, in the Sham protocol, six participants answered that they did not feel itching (level 0) and six participants answered that they felt itching with soft intensity (level 1) and in the CT10, AiC protocol, eight participants reported that they did not feel itching (level 0), two participants reported that they felt itching softly (level 1), one participant responded that he/she felt itching moderately (level 2) and one participant answered that he/she felt itching intensely (Level 3). For the duration of the electrical stimulation, level 0 indicates that the subject did not feel the electrical stimulation, and levels 1, 2, 3, and 4 indicate that the participant felt a very short, short, long, or very long duration electrical stimulation, respectively. The white dots inside the geometric shapes correspond to the medians of the results in each condition.

electrodes positioned on the back and on the iliac crests. Two subjects developed small blisters on the skin in the region in contact with the stimulation electrodes which healed in a few days. One subject presented hypersensitivity to the electrical stimulation, canceling the participation in the experiment.

The raw COP data and the electrical stimulation assessment are available in an online open access repository [84].

## Discussion

Previous studies of tsDCS have shown that DC electrical stimulation can induce motor performance improvements in tasks executed with the lower limb in healthy young subjects [21, 22]. Other studies reported that tsDCS can modify electrophysiological responses related to the spinal cord as well as to supramedulary neural pathways [1, 3–6, 13, 18]. However our results showed that tsDCS caused no statistically significant effect on postural sway during quiet standing of healthy young individuals. The hypothesis of this work was based on published experimental data suggesting that both segmental and suprasegmental regions of the central nervous system are involved in postural control during upright posture [47, 48, 62, 64]. In particular, DC stimulation in the vinicity of T10 has been reported to modulate the behavior of spinal and/or supramedullary neuronal networks [12, 17–20], suggesting that tsDCS around the lumbar level could be effective in influencing postural control and possibly also postural sway.

In our experimental protocol, to mitigate possible cutaneous sensory cues produced by the electrical stimulation on the iliac crests, improving experimental blinding, we selected stimulation electrodes with surface area of 50 cm$^2$. To keep the current density over the spinal cord region similar to the usually adopted in other tsDCS studies, an electrical stimulation intensity of 5 mA was selected. This electrical stimulation configuration was slightly different from the generally used (35 cm$^2$ electrodes and stimulation intensity of 2.5 mA) [1, 4, 8, 9, 12, 17, 71, 72]. The relationship between current intensity and observed effects may be non-linear in tsDCS protocols [85–89]. An increase of the electrical stimulation current could result in the reduction or cancellation of the neuromodulation effects [90]. It is possible that although the use of larger surface electrodes may have reduced skin sensory cues, the use of a higher current intensity in our experimental protocol may have resulted in an attenuation of the effects of the electrical stimulation on neural pathways related to quiet standing postural control.

Recent computational studies of tsDCS have reported that the placement of electrodes over T10 and over the iliac crest optimizes the stimulation of the region of the lumbosacral enlargement [24, 27], which contains a large amount of axonal and dendritic segments as well as a great number of synapses related to lower limb activation [91, 92] and thus certainly related to postural control during standing. In addition, computer simulation of an approximate mathematical model developed in our laboratory, showed that the use of a bilateral placement of electrodes on the iliac crests allowed a reduction in the electric current density in this region of the skin, when compared to the unilateral placement, without modifying the electric field generated in the spinal cord [93]. These evidences motivated us to use an electrode positioned over T10 and two electrodes over the iliac crests bilaterally. This configuration is slightly different from the one generally employed in previous studies that commonly use one electrode positioned over T10/T11 and the second electrode over the upper torso region [1, 3–13, 17, 71, 94]. One cannot rule out the possibility that the adopted electrode positions generated an electric field in the spinal cord with inappropriate orientation to induce sufficient membrane potential changes of axons and dendrites of tracts and neurons involved in postural control during stance. The orientation of the electrical field with respect to the neuronal

compartments influence the level of polarization/depolarization generated by a DC stimulation [24, 95]. In a study that used electrode placements rather similar to the adopted in the present study, albeit with a smaller current intensity (2.5 mA), no significant changes of some electrophysiological responses (F-Wave, H-reflex, MEP) associated with the lower limbs were found [73].

Previous studies of tsDCS adopted for the Sham protocol either a short-duration initial electrical stimulation [1, 4, 6, 8, 10–13, 72, 73, 94, 96–98] or repetitive pulsed electrical stimulation [3, 7] to mimic the cutaneous sensations elicited by the tsDCS. In preliminary tests performed in our laboratory we noticed that these techniques were far from being effective in masking the absence of electrical stimulation during the Sham protocol. We also observed that it was possible to easily identify the difference between the cutaneous sensory experience of different electrical protocols (C-T10, CiC) and (A-C10, CiC) even with the commonly used electrical current intensity of 2.5 mA and 35 cm$^2$ electrodes. In addition, there seems to be a lack of data on minimum limits in duration and intensity of the DC stimulation that could avoid some long-lasting neuromodulation on the spinal circuitry caused by the sham stimulation. Thus it cannot be guaranteed that even a relatively short duration stimulation could not result in some unwanted effect that could influence postural sway. In order to try to mitigate these problems, we developed a novel initialization of the electrical stimulation for all electrical stimulation protocols (Fig 1, upper right plot). The developed startup electrical stimulation minimizes differences of cutaneous cues produced by different protocols as the same polarity sequence is applied to the electrodes in the initial instants of all protocols, hence improving the experimental blinding. The bipolar aspect of the waveform of this novel stimulation startup also results in a zero total electric charge applied to the spinal cord, minimizing unwanted effects of even a short duration electrical stimulation of the Sham protocol [74]. No statistically significant differences were found between the results of the adopted perceptual assessment of the different stimulation protocols indicating that the proposed stimulation methodology could be useful in future work employing tsDCS. Although common in tDCS studies, to our knowledge, no previous tsDCS studies have conducted a quantitative assessment of the experimental blinding. Hence, the proposed methodology for assessing the experimental blinding could be adopted in future studies to evaluate or compare different sham/blinding protocols in tsDCS.

Although statistically significant differences were observed in the parameters $f50p\_PSD_{AP}$ and $\alpha L_{ML}$, they were not related to the electrical stimulation. For the parameter $f50p\_PSD_{AP}$ the change was observed even before the application of electrical stimulation, indicating that for this parameter the individuals presented different behavior from the COP baseline on different days. The difference observed in parameter $\alpha L_{ML}$ is not related to the interaction between stimulation protocol and stimulation stage, therefore excluding effects induced by the electrical stimulation. These differences can be attributed to uncontrolled factors in the experimental protocol, such as the effects of sleep [99], anxiety [100], and circadian cycle [101] on postural control of quiet upright standing.

In Fig 3 it can be noted that for some parameters, such as $AREAE$, $CI_{AP}$, $f50p\_PSD_{AP}$, $\Delta TC_{AP}$ and $SD_{AP}$, results of some subjects could be qualitatively considered outliers. However, no reasons were found for the exclusion of these data from the final analysis by the meticulous visual inspection of the COP data in the AP direction and the observed participant's behavior during the data acquisition. In addition, the subjects that could be considered outliers are different for different parameters. Thus, we chose not to exclude these data to avoid biasing the final results [83]. Great care was taken to standardize experimental conditions, such as keeping the same state of alertness, positioning of the feet on the force platform, laboratory temperature, and minimizing sensory, auditory, and visual cues, to reduce the influence of factors other than those produced by the electrical stimulation protocol on the postural control [102–

104]. In addition, we followed all the relevant procedures for ensuring the metrological quality of the results, such as using calibrated instruments and ensuring the correct timing and intensity of the electrical current applied by the tsDCS protocol.

It could be that the choice of a group of healthy young people in this study may have hindered changes of COP parameters by tsDCS as this group might not benefit from the neuromodulation produced by electrical stimulation. However, some studies have shown that modifying the proprioceptive feedback during quiet standing, even in young and healthy individuals, can improve the performance of this task [32, 105, 106]. As we found no previous research on the putative effects of tsDCS on postural oscillations in humans it seemed reasonable to depart from a control sample of young and healthy subjects so that many issues related to the stimulation technique (blinding protocol, safety, tolerability and efficacy of the methodology) could be developed and analysed. However, an effort was made to increase the difficulty of the quiet standing task, as visual and auditory cues were suppressed by using opaque glasses and headphones while the task was being performed.

Although no statistically significant changes in postural sway were induced by different polarizations of tsDCS in healthy young subjects, it cannot be stated that tsDCS did not induce any postural control changes in this group during quiet standing. It is possible that the adopted stimulation could be affecting other aspects of leg motor control not measured in the present experiment, for example, reactions to a sudden external perturbation [107]. In terms of future investigations, keeping postural sway as an indicator of standing motor control, or focusing on some other aspect of leg motor control, one could choose different electrical stimulation settings, such as a 2.5 mA DC electrical stimulation and/or electrode placement at T10 and at the right shoulder. Future research could verify if tsDCS can induce changes in quantifiers of postural control in other groups of subjects such as elderly, elderly with history of falls or individuals with some specific neurological disorder.

In a relatively new field of research, as is the case of tsDCS, it is important that negative results also be published to provide a wider view of the technical and physiological issues involved, helping direct progress in future research. The present study contributes to the field of tsDCS knowledge by presenting the non-observation of effects of tsDCS on postural sway during stance for a specific experimental protocol applied to healthy young subjects, but also proposing an alternative stimulation protocol that addresses the issue of blinding the subject to the different stimulus conditions including the sham trials. Finally, the present results open up the way for further research on tsDCS both in terms of methodology and the specificities of the subset of subjects to be tested with this technique.

## Supporting information

**S1 File. Electrical stimulation assessment.** Translated version of electrical stimulation protocol the assessment form.
(DOCX)

**S2 File. Supplementary information.** Tables of averages and standard deviations of all COP parameters at different stages (t0, t1 and t2) of the three electrical stimulation protocols, quartile ranges of the electrical stimulation assessments answers and results of the statistical analysis both for the COP parameters and for the results of the assessment of electrical stimulation.
(PDF)

## Author Contributions

**Conceptualization:** Felipe Fava de Lima, Cristiano Rocha Silva, Andre Fabio Kohn.

**Data curation:** Felipe Fava de Lima.

**Formal analysis:** Felipe Fava de Lima, Andre Fabio Kohn.

**Funding acquisition:** Andre Fabio Kohn.

**Investigation:** Felipe Fava de Lima, Cristiano Rocha Silva, Andre Fabio Kohn.

**Methodology:** Felipe Fava de Lima, Cristiano Rocha Silva, Andre Fabio Kohn.

**Project administration:** Felipe Fava de Lima, Andre Fabio Kohn.

**Resources:** Andre Fabio Kohn.

**Software:** Felipe Fava de Lima.

**Supervision:** Andre Fabio Kohn.

**Validation:** Felipe Fava de Lima.

**Visualization:** Felipe Fava de Lima.

**Writing – original draft:** Felipe Fava de Lima, Andre Fabio Kohn.

**Writing – review & editing:** Felipe Fava de Lima, Cristiano Rocha Silva, Andre Fabio Kohn.

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
