## [Decision Letter · Decision Letter 0]

23 Dec 2021

PONE-D-21-37330Transcutaneous spinal direct current stimulation (tsDCS) does not affect postural sway during quiet upright standingPLOS ONE

Dear Dr. Fava de Lima,

Thank you for submitting your manuscript to PLOS ONE. After careful consideration, we feel that it has merit but does not fully meet PLOS ONE’s publication criteria as it currently stands. Therefore, we invite you to submit a revised version of the manuscript that addresses the points raised during the review process. As you will see below, the Reviewers were generally positive regarding your manuscript. While Reviewer #1 has only minor issues to address, Reviewer #2 raises some significant concerns regarding the rationale for the study and some methodological aspects (e.g., description of participants, clarifications for excluding data). Please ensure that all the concerns are properly addressed in the revised version.

We look forward to receiving your revised manuscript.

Kind regards,

François Tremblay, PhD

Academic Editor

PLOS ONE

Journal Requirements:

"This research was supported by Grants provided by the Foundation for Research 368

Support of the State of S˜ao Paulo (FAPESP 2016/10614-4) and by the Brazilian 369

Council of Science and Technology (CNPq, 303809/2016-7)."

"CRS -  Foundation for Research

Support of the State of São Paulo (FAPESP 2016/10614-4) (https://fapesp.br/)

AFK -  Brazilian

Council of Science and Technology (CNPq, 303809/2016-7) (https://www.gov.br/cnpq/pt-br)

4. Please include your tables as part of your main manuscript and remove the individual files. Please note that supplementary tables (should remain/ be uploaded) as separate "supporting information" files

Reviewers' comments:

Reviewer's Responses to Questions

**Comments to the Author**

1. Is the manuscript technically sound, and do the data support the conclusions?

Reviewer #1: Yes

Reviewer #2: Partly

2. Has the statistical analysis been performed appropriately and rigorously? 

Reviewer #1: Yes

Reviewer #2: Yes

3. Have the authors made all data underlying the findings in their manuscript fully available?

Reviewer #1: Yes

Reviewer #2: Yes

4. Is the manuscript presented in an intelligible fashion and written in standard English?

Reviewer #1: Yes

Reviewer #2: Yes

5. Review Comments to the Author

Reviewer #1: The paper by de Lima and co-workers reports interesting findings about the possibility to modulate postural sway by using spinal tDCS. The methods are sound and the paper requires only minor revisions:

- The Introduction needs a language editing;

- Did Authors use a questionnaire to assess possible side-effects of tsDCS?

- Though do not statistically significant, the results may depend on the montage used and probably differ between health and disease. The uthors should discuss both these critical points in detail.

- Which are the putative supraspinal pathways modulated by this particular montage?

Reviewer #2: The authors present their findings related to the change in static balance performance in conjunction with transcutaneous direct current spinal stimulation. This investigation was performed in a young healthy sample and while there report negative findings the authors have proposed a novel sham method. One factor that reduces my enthusiasm for the manuscript is that the authors do not report how they determined their sham was as and/or more effective than traditional tsDCS methods. Adding this information would increase the beneficial information disseminated and help other investigators perform more rigorous future tsDCS studies. To strengthen the manuscript's importance to the greater body of research the authors should also better justify their sample selection. Typically the young healthy population performs well on tests of static balance, making this population a less than ideal population to sample from. However, because there are not many tsDCS studies one could better justify this sample by including tests of safety, tolerability, etc. Another point that could be better justified is the eyes closed methodology. This was likely done to increase the difficulty of the balance test in turn leading to a greater probability of detecting an effect.

In general, I believe the manuscript possesses merit for publication but believe it can be significantly strengthened by addressing the above comments. A list of specific comments is presented below.

Title and Abstract: I suggest characterizing the sample as young and healthy in the title and abstract. Readers will likely acknowledge that the lack of tsDCS effect is because of the sample and not because the technology doesn’t work.

Introduction:

Line 70: Your null hypothesis is misstated

Methods:

L117: Report the range of stimulation/current densities in the literature with references. This will allow readers to understand where/how your protocol fits into what has previously been done.

Report the balance of the groups. With a small sample, randomization may lead to an imbalance in the order of the performed simulations. Is there an order effect because more sham procedures were done before actual stimulation trials?

Better describe why some data was excluded. The figure provided shows an example of excluded data but explain to the reader why that pattern is abnormal. Provide references to justify if possible.

Better describe the identification of T10, I assume it was through palpation? Was it supervised by physical therapists or other clinicians? If the research staff is properly trained or has experience please cite a reference.

Results:

L214: Report Ht and Wt. BMI reflects health status and is not relevant to this investigation.

L225: Null hypothesis doesn’t match the intro, ensure consistency (intro null hypothesis is not correct)

L230: Indicate where in the figure this significant finding is. Added letters or numbers to each panel of the figure to allow more clear references between the text and figure.

Report if your novel blinding technique worked. How many participants were able to correctly identify if they received active or sham stimulation.

Report all minor and major adverse events. This will help establish the safety of the technique and add importance to your negative findings.

Discussion:

Line 241-252: It is unclear why AC is mentioned. There is no information about AC in the intro. The discussion would be more clear if you restated your hypotheses and findings.

L320: No data presented to support your blinding method

L342: You present no effect sized in your results and you did not say you calculated them in your methods. Either remove from your discussion or add the information to the previous sections. If you discuss data from the study it should be reported in the appropriate sections. (it can be supplementary data if you so choose)

You report several significant interactions but there is no mention of them in your discussion. You cannot ignore findings that you report even if they may not have much clinical relevance. Discuss why/what those findings may or may not signify

Discuss how much of an effect you might have found in young healthy individuals. Cite previous works showing young healthy balance can be improved and why this was the proper population to test your hypotheses.

Figure 3: Added identifiers to each panel for easy reference from the test. You state significant results but did not indicate them in the figure or figure legend (F50p_PSD…)

6. PLOS authors have the option to publish the peer review history of their article (what does this mean?). If published, this will include your full peer review and any attached files.

Reviewer #1: **Yes: **Tommaso Bocci

Reviewer #2: **Yes: **John H. Kindred

---

## [Author Response · Author response to Decision Letter 0]

22 Mar 2022

Journal Requirements:

Authors: Manuscript meets PLOS ONE’s style requirements.

"This research was supported by Grants provided by the Foundation for Research 368

Support of the State of S˜ao Paulo (FAPESP 2016/10614-4) and by the Brazilian 369

Council of Science and Technology (CNPq, 303809/2016-7)."

"CRS -  Foundation for Research

Support of the State of São Paulo (FAPESP 2016/10614-4) (https://fapesp.br/)

AFK -  Brazilian

Council of Science and Technology (CNPq, 303809/2016-7) (https://www.gov.br/cnpq/pt-br)

Authors: Funding information was removed from manuscript’s Acknowledgments Section. Update the Funding Statement above as following:

São Paulo Research Foundation (FAPESP grant #2016/10614-4 from CRS) (https://fapesp.br/)

Brazilian Council of Science and Technology (CNPq grant #303809/2016-7 from AFK) (https://www.gov.br/cnpq/pt-br)

Authors: The data set used in this study will be available after acceptance

of the manuscript for publication at OSF public repository (DOI 10.17605/OSF.IO/C24TA on https://osf.io/c24ta/). 

4. Please include your tables as part of your main manuscript and remove the individual files. Please note that supplementary tables (should remain/ be uploaded) as separate "supporting information" files

Authors: Supplementary table will be uploaded as supporting information files.

Review Comments to the Author

Reviewer #1 

The paper by de Lima and co-workers reports interesting findings about the possibility to modulate postural sway by using spinal tDCS. The methods are sound and the paper requires only minor revisions:

Authors: We thank the reviewer for the revision of the paper and the useful suggestions. We think the revised version improved considerably due to the valuable inputs provided by thereviewer.

- The Introduction needs a language editing;

Authors: The text of the entire manuscript was revised.

- Did Authors use a questionnaire to assess possible side-effects of tsDCS?

Authors: The results of the electrical stimulation evaluation questionnaire have been added to the manuscript. Side-effects information has also been added to the text (L189 and L279).

- Though do not statistically significant, the results may depend on the montage used and probably differ between health and disease. The authors should discuss both these critical points in detail.

Authors:The manuscript discusses the potentials of further studies in different populations (the elderly, elderly with a history of falls, and neuropathic individuals) as well as the employment of different electrode positions and/or current intensities (L403).

- Which are the putative supraspinal pathways modulated by this particular montage?

Authors: The manuscript has been edited with the addition of references which report effects in supramedullary pathways and nuclei due to DC stimulation around T10 used in our work (please see lines 14-19 and 299-302). 

Reviewer #2: 

The authors present their findings related to the change in static balance performance in conjunction with transcutaneous direct current spinal stimulation. This investigation was performed in a young healthy sample and while there report negative findings the authors have proposed a novel sham method. One factor that reduces my enthusiasm for the manuscript is that the authors do not report how they determined their sham was as and/or more effective than traditional tsDCS methods. Adding this information would increase the beneficial information disseminated and help other investigators perform more rigorous future tsDCS studies. To strengthen the manuscript's importance to the greater body of research the authors should also better justify their sample selection. Typically the young healthy population performs well on tests of static balance, making this population a less than ideal population to sample from. However, because there are not many tsDCS studies one could better justify this sample by including tests of safety, tolerability, etc. Another point that could be better justified is the eyes closed methodology. This was likely done to increase the difficulty of the balance test in turn leading to a greater probability of detecting an effect.

In general, I believe the manuscript possesses merit for publication but believe it can be significantly strengthened by addressing the above comments. A list of specific comments is presented below.

Authors: We thank the reviewer for the many useful suggestions which were valuable in guiding our efforts to improve the manuscript.

Title and Abstract: I suggest characterizing the sample as young and healthy in the title and abstract. Readers will likely acknowledge that the lack of tsDCS effect is because of the sample and not because the technology doesn’t work.

Authors: The title and abstract of the manuscript have been modified to highlight that the results of the study were obtained from healthy young participants.

Introduction:

Line 70: Your null hypothesis is misstated

Authors: The introduction (L84) has been changed to correctly state the hypothesis of the study.

Methods:

L117: Report the range of stimulation/current densities in the literature with references. This will allow readers to understand where/how your protocol fits into what has previously been done.

Authors: Stimulation current densities in the literature with references were added in the methods allowing the comparison between the stimulation parameters adopted and other studies in the literature (L128).

Report the balance of the groups. With a small sample, randomization may lead to an imbalance in the order of the performed simulations. Is there an order effect because more sham procedures were done before actual stimulation trials?

Authors:The balance of sequences of the electrical stimulation protocol is now available in the Supplementary Information File in the Additional Information Files. The text of the manuscript has been modified (L251) to emphasize that the sequences of the electrical stimulation protocols were selected randomly, but in a way that prioritized the balance of sequences across participants.

Better describe why some data was excluded. The figure provided shows an example of excluded data but explain to the reader why that pattern is abnormal. Provide references to justify if possible.

Authors: Some data were excluded because they presented instants when participants performed some undesirable movement (distraction, deep breathing and small arm movements) that could interfere the COP parameters results. The text of the methodology has been changed to better describe the data exclusion(L211) .

Better describe the identification of T10, I assume it was through palpation? Was it supervised by physical therapists or other clinicians? If the research staff is properly trained or has experience please cite a reference.

Authors: The identification of T10 was obtained by palpation. The methodology section has been changed (L125) to better describe the identification method used. One of the authors (Dr. Cristiano Rocha Silva), also responsible for data collection, is a physical therapist and supervised the identification procedure of the intervertebral spaces of the participants for the correct positioning of the electrodes.

Results:

L214: Report Ht and Wt. BMI reflects health status and is not relevant to this investigation.

Authors:The text has been amended (L251) with the weight and height information of the subjects.

L225: Null hypothesis doesn’t match the intro, ensure consistency (intro null hypothesis is not correct)

Authors:The introduction section of the manuscript has been modified (L84) in order to define the hypothesis of the paper correctly.

L230: Indicate where in the figure this significant finding is. Added letters or numbers to each panel of the figure to allow more clear references between the text and figure.

Authors: Indications of significant statistical differences were added to Figure 3 and letters were used to identify each panel.

Report if your novel blinding technique worked. How many participants were able to correctly identify if they received active or sham stimulation.

Authors:The result of the statistical test applied in the assessments of the electrical stimulation was added to the manuscript (L189 and L279) .The result indicates that no statistically significant differences were observed between the responses provided by the participants in different electrical stimulation protocols, suggesting the effectiveness of the experimental blinding of the proposed methodology. 

Report all minor and major adverse events. This will help establish the safety of the technique and add importance to your negative findings.

Authors: A statement of minor and major adverse effects has been added to the text (L282) . 

Discussion:

Line 241-252: It is unclear why AC is mentioned. There is no information about AC in the intro. The discussion would be more clear if you restated your hypotheses and findings.

Authors: The text involving AC stimulation was removed since it could confuse the understanding of the paragraph and of the methodology employed in the study. The text has been changed (L291) to emphasize the relevance of the experiment.

L320: No data presented to support your blinding method

Authors: The results of the electrical stimulation assessment were added to the manuscript and the text of the discussion was changed (L358) to give a better support for the features of the proposed electrical stimulation method.

L342: You present no effect sized in your results and you did not say you calculated them in your methods. Either remove from your discussion or add the information to the previous sections. If you discuss data from the study it should be reported in the appropriate sections. (it can be supplementary data if you so choose)

Authors: Effect sizes were removed from the discussion.

You report several significant interactions but there is no mention of them in your discussion. You cannot ignore findings that you report even if they may not have much clinical relevance. Discuss why/what those findings may or may not signify

Authors: A paragraph has been added discussing the significant iterations found (L366) .

Discuss how much of an effect you might have found in young healthy individuals. Cite previous works showing young healthy balance can be improved and why this was the proper population to test your hypotheses.

Authors: A discussion with references to previous studies showing the improvement of balance in healthy young people has been added to the text (L403). 

Figure 3: Added identifiers to each panel for easy reference from the test. You state significant results but did not indicate them in the figure or figure legend (F50p_PSD…)

Authors: Indications of significant statistical differences were added to Figure 3 and letters were used to identify each panel.

---

## [Decision Letter · Decision Letter 1]

14 Apr 2022

Transcutaneous spinal direct current stimulation (tsDCS) does not affect postural sway of young and healthy subjects during quiet upright standing

PONE-D-21-37330R1

Dear Dr. Fava de Lima,

We’re pleased to inform you that your manuscript has been judged scientifically suitable for publication and will be formally accepted for publication once it meets all outstanding technical requirements.

Kind regards,

François Tremblay, PhD

Academic Editor

PLOS ONE

Additional Editor Comments (optional):

Please make sure that the manuscript is properly edited for grammatical and typographical errors when preparing the final version.

Reviewers' comments:

Reviewer's Responses to Questions

**Comments to the Author**

1. If the authors have adequately addressed your comments raised in a previous round of review and you feel that this manuscript is now acceptable for publication, you may indicate that here to bypass the “Comments to the Author” section, enter your conflict of interest statement in the “Confidential to Editor” section, and submit your "Accept" recommendation.

Reviewer #2: All comments have been addressed

2. Is the manuscript technically sound, and do the data support the conclusions?

Reviewer #2: Yes

3. Has the statistical analysis been performed appropriately and rigorously? 

Reviewer #2: Yes

4. Have the authors made all data underlying the findings in their manuscript fully available?

Reviewer #2: Yes

5. Is the manuscript presented in an intelligible fashion and written in standard English?

Reviewer #2: Yes

6. Review Comments to the Author

Reviewer #2: I thank the authors for putting in the work to improve their submission. The authors have addressed all of my concerns and I look forward to seeing more of their work in the future.

7. PLOS authors have the option to publish the peer review history of their article (what does this mean?). If published, this will include your full peer review and any attached files.

Reviewer #2: **Yes: **John H. Kindred, Ph.D.

---

## [Editor Report · Acceptance letter]

20 Apr 2022

PONE-D-21-37330R1 

Transcutaneous spinal direct current stimulation (tsDCS) does not affect postural sway of young and healthy subjects during quiet upright standing 

Dear Dr. Fava de Lima:

I'm pleased to inform you that your manuscript has been deemed suitable for publication in PLOS ONE. Congratulations! Your manuscript is now with our production department. 

Kind regards, 

on behalf of

Dr. François Tremblay 

Academic Editor

PLOS ONE